# Association between Serum Total and Specific Immunoglobulin E Levels and Body Height: A Cross-Sectional Study of Children and Adolescents

**DOI:** 10.3390/children9050661

**Published:** 2022-05-04

**Authors:** Ling-Sai Chang, Jou-Hui Li, Pei-Ming Wang, Chih-Fang Huang, Ho-Chang Kuo

**Affiliations:** 1Department of Pediatrics, Chang Gung Memorial Hospital, Kaohsiung 83301, Taiwan; joycejohnsyoko@gmail.com; 2College of Medicine, Chang Gung University, Taoyuan City 33302, Taiwan; 3Department of Family Medicine, Chang Gung Memorial Hospital, Kaohsiung 83301, Taiwan; sabrina617111@hotmail.com (J.-H.L.); sel7129@adm.cgmh.org.tw (P.-M.W.); pauline.chifang@gmail.com (C.-F.H.); 4Department of Respiratory Therapy, Kaohsiung Chang Gung Memorial Hospital, Kaohsiung 83301, Taiwan

**Keywords:** allergy, body height, body weight, childhood, total and specific IgE

## Abstract

Background: The atopy rate in children has increased significantly. Atopy and growth are connected in a multifactorial manner and are important health issues for children around the world. The principal research question in this cross-sectional investigation concerned the association between serum total, specific immunoglobulin E (IgE) levels, and body height (BH)/weight (BW)/body mass index (BMI). Methods: A total of 993 subjects were enrolled for analysis retrospectively with allergic diseases and aged from 6 months to 18 years during the years 2015–2016. A complete panel of 36 allergen-specific IgE was taken from each participant using the MAST allergen test as well as their BH, BW, BMI, and total IgE levels. Results: There was a statistically significant positive association between the total IgE levels with BH (N = 348) and BW (N = 623) in the preschool age group (<6 years old, *p*-values of 0.009 and 0.034, respectively). In the preschool group, the total IgE levels showed a positive association with house dust (*p* < 0.001), cockroach mix (*p* < 0.001), Dermatophagoides farina (*p* < 0.001), and Dermatophagoides pteronyssinus (*p* < 0.001). After performing a general linear model followed by a backward selection of variables with age, sex, specific IgE, and total IgE, egg white sensitization demonstrated a significant negative association with BH (*p* = 0.009), and Dermatophagoides farina sensitization showed a significant positive association with BH (*p* = 0.006). The analysis showed that, in this model, the level of total IgE was not associated with BH. Conclusions: The results of this study indicate that the level of total IgE was not associated with BH in the preschool age group. Future studies are needed to replicate the results in outcome with follow-up allergic cohorts.

## 1. Introduction

The atopy rate in children has increased significantly in some areas, particularly in industrialized countries [1,2,3,4]. The most representative data on these divergent time trends come from the ISAAC project, showing an increased prevalence of allergic rhinoconjunctivitis and eczema symptoms in the 6–7 and 13–14 year age group in Taiwan after seven years, the first survey being given between 2002 and 2003 [4]. According to data from studies done in Taiwan, the prevalence of childhood asthma increased from 1.3% in 1974 to 5.1% in just ten years [5]. A similar study in Taiwan demonstrated that atopy has continued to increase in recent decades. For example, the prevalence of asthma more than tripled from 1987 to 2002, jumping from 2.2% to 7.0%. Such increases were also noted in allergic rhinitis (AR, from 5.1% in 1987 to 27.6% in 2002) and atopic dermatitis (AD, from 1.1% in 1987 to 3.4% in 2002) [6].

Allergies are increasingly common these days [7,8] and tend to produce higher levels of immunoglobulin (Ig) E in response to contact with allergens. To diagnose allergic disorders in individuals with symptoms, allergen-specific IgE, which can recognize specific allergens, is a vital tool [9,10]. Atopy is generally accepted to potentially impact growth conditions in children [11], but high-quality evidence is still lacking. Young adults with IgE-mediated cow’s milk allergy from infancy have been shown to have significantly lower body height (BH) compared to controls [12]. Various factors can influence growth in children, and dietary control with allergen avoidance in food allergy patients has been indicated as a contributing factor to growth parameters [13].

Children with allergic diseases, particularly AD or AR, frequently experience sleep disturbances [14,15]. For example, among atopic children, the airflow restriction caused by allergic inflammation or obesity may result in sleep disorders [16]. While sleep disorders are believed to impact growth in atopic children, a conclusion has not yet been reached.

Allergic diseases and growth, which interrelate reciprocally, remain an area of great concern [17]. In this cross-sectional study, we aimed to investigate the association between total, specific IgE and BH/body weight (BW)/body mass index (BMI). We compared the total and specific IgE levels among allergic children with different BH/BW/BMI levels and hypothesized that a lower BH was not associated with having higher IgE levels. Our next research question focused on the effect of the age of the allergic children on the association.

## 2. Methods

### 2.1. Participants and Study Design

The participants enrolled in our cross-sectional study were children with a current diagnosis of an atopic status, including AD, AR, and/or asthma, given by a doctor in the outpatient pediatrics department at Kaohsiung Chang Gung Memorial Hospital in Taiwan. We enrolled a total of 993 subjects with ages from 6 months to 18 years during December 2015–August 2016 in order to determine the relationship between IgE levels and body size, including BH, BW, and BMI. For consideration of the pubertal effect, we divided the data into groups according to the definition of precocious puberty age in Taiwan, which is 11 years old for boys and 10 years old for girls [18]. Children with an underlying medical condition or congenital abnormality were excluded, as were subjects with heart, thyroid, chronic lung, renal, or liver disease. Informed consent was waived, and all methods were conducted in accordance with the relevant guidelines that were approved by the Institutional Review Board of Chang Gung Medical Foundation (IRB No: 104-A096B).

### 2.2. Assessments

A complete panel of 36 allergen-specific IgE was taken from each participant’s peripheral blood by using the MAST allergen test (the multiple allergosorbent test system, chemiluminescent immunoassay sandwich method), BW (in kilograms), BH (in centimeters), and the total IgE data. Of the 993 participants, 581 (58.5%) had a BH measured using appropriate measuring devices. The missing data for BH were not included in the analysis related to BH or BMI. The study size provided for a power greater than 90%. Standing BH was measured without shoes. The serum total IgE levels and 36 allergen-specific IgE antibodies against latex, chicken feathers, Bermuda grass, black willow, eucalyptus, Japanese cedar, white mulberry, pigweed, ragweed mix, timothy grass, Alternaria, aspergillus, Cladosporium, penicillium, cat, dog, house dust, cockroach mix, Dermatophagoides farina, Dermatophagoides pteronyssinus, avocado, pork, beef, milk, cheese, shrimp, crab, clam, codfish, tuna, peanut, soy, wheat, brewer’s yeast, egg yolk, and egg white were analyzed along with BH and BW. The growth status was evaluated according to the “child growth standards” of the World Health Organization (WHO) (https://www.who.int/toolkits/ child-growth-standards) and (https://growth.healthinfo.tw/) during the study period from December 2015 to August 2016.

### 2.3. Statistical Methods

We used the statistical software SPSS (Version 13.0) to perform all analyses. Participants were categorized into three different age groups, including preschool (<6 years old), elementary school (6–12 years old), and high school (12–18 years old). We evaluated the statistical significance between body size and total IgE level using Pearson’s correlation and the statistical significance between body size and allergen-specific IgE (grade zero to four) using Spearman’s correlation [19]. A *p*-value less than 0.05 was considered statistically significant for the main effects. We determined the relationship between a specific IgE and body size by applying a general linear model followed by a backward selection of variables including age, sex, specific IgE, and total IgE, taking into account that BH increased with age and that age, total IgE, and sex had significant associations with allergen sensitization [20,21]. Age and total IgE were considered to be continuous variables. Sex and specific IgE were considered to be categorical variables. Grade zero was designated as negative sensitization. Grades one to four were chosen to refer to positive sensitization [22]. Then, a false discovery rate correction for multiple comparisons was conducted. The statistical power was calculated in the general linear model.

### 2.4. Patient and Public Involvement

Patients and/or the public were not involved in the designing, conducting, or the dissemination plans of this research.

## 3. Results

Preschool subjects accounted for more than 62.7% (623 cases) of subjects in this study, while 27.1% (269 cases) were elementary school students, and 10.2% (101 cases) were high school students. In addition, 60.2% (598 cases) were male. Body weight was measured in all 993 allergic children, but 412 children had no height measurement. The WHO child growth standards showed BH (43.09 ± 29.9 %, N = 581), BW (45.81 ± 32.29, N = 993), and BMI (48.12 ± 33.67, N = 581). The median levels of serum total IgE were 443 kU/L (range, 2–9756 kU/L). As shown in Table 1, approximately half of the cases had a sensitivity to dust mites, which agreed with previous studies that have suggested that this is the most common sensitizer in Taiwan [23,24]. Most subjects showed a sensitivity to Dermatophagoides farina (50.3%) and Dermatophagoides pteronyssinus (46.2%), followed by house dust (27.5%) and cockroach mix (21.3%). Sensitivities to white mulberry and ragweed mix were the least common. The case numbers were significantly higher among subjects allergic to aeroallergens than in those with food allergens.

There were statistically significant positive associations between total IgE levels with BH (N = 581, *p* < 0.001) and BW (N = 993, *p* = 0.007) across all age groups. When divided into the different age groups, statistically significant positive associations between BH (Pearson’s correlation coefficient *r* = 0.139, N = 348, *p* = 0.009) and BW (Pearson’s correlation coefficient *r* = 0.085, N = 623, *p* = 0.034) with total IgE were found only in the preschool age group (Table 2). However, as can be seen from the coefficients of *r* < 0.2, we did not find a clinically relevant association in Table 2 [19]. No significant differences were found between the total IgE and BH in either the elementary school group (N = 168, *p* = 0.946) or the high school group (N = 65, *p* = 0.715). Likewise, no significant differences were found between the total IgE and BW (N = 269 and 101, respectively; *p*-value = 0.894 and 0.680, respectively). The results of this study were adjusted according to the average pubertal age; pubertal status showed no significant effect on the total IgE and BH/BW/BMI. BH showed a significant positive association with house dust, cockroach mix, Dermatophagoides farina, and Dermatophagoides pteronyssinus (all *p* < 0.001) but a negative association with egg whites (*p* < 0.001) (Table 3). Similar results were found for BW (Table 4). When BMI was expressed as a percentile, we found no correlation between the total IgE and the BMI percentile (*p* = 0.615).

In the preschool group, BH showed a significant positive association with house dust (*p* < 0.001), cockroach mix (*p* < 0.001), Dermatophagoides farina (*p* < 0.001), and Dermatophagoides pteronyssinus (*p* < 0.001). Table 3 and Table 4 show significant associations with *p* < 0.05 and coefficients *r* > 0.2 or <−0.2 between specific IgE and BH/BW. Similar results were found with BW in the preschool group. Of the 36 kinds of specific IgE, it was found that egg whites demonstrated a significant negative association with BH in subjects under 6 years old after we performed a general linear model (*p* = 0.009) with a power greater than 90%. Dermatophagoides farina sensitization showed a significant positive association with BH (*p* = 0.006). The analysis showed the level of total IgE was not associated with BH in this model.

## 4. Discussion

We identified BH as having a positive association with Dermatophagoides farina sensitization in the preschool subgroup of allergic children. The positive association between Dermatophagoides farina sensitization and BH may possibly be due to an increased surface area in contact with the allergens. Egg white sensitization showed a statistically significant negative association with BH. The impact of BH was similar to that in a BH survey of young adults with another common IgE-mediated food allergy, cow milk [12]. However, this phenomenon disappeared after school. It may be because the diet control before school is stricter, which affects nutrient intake. Future studies are needed to replicate the results in outcome with a follow-up of allergic cohorts.

The prevalence of allergic diseases has increased in different countries in the past decade [25]. The data from the 2016 National Health Interview Survey (NHIS) showed that 12.8% of children and adolescents had asthma [26]. Growing evidence has indicated that total IgE measurements may be adopted to help in the diagnosing of atopic diseases. Elevated levels of total serum IgE are often observed in allergic patients [27]; approximately 4% of US children have an IgE-mediated food allergy [28]. Increased IgE levels have been most consistently associated with AD, followed by atopic asthma, then AR [29]. As much as 80% to 85% of patients with AD have been observed to have elevated total IgE levels [30]. The increased total and allergen-specific IgE level in allergic patients associated with sensitization is also evident but less pronounced in disease severity [31]. Nevertheless, other diseases such as parasitic infestation, malignancy, immunodeficiencies, and drug effects may also cause elevated IgE levels [29]. The steady trend of IgE levels increasing with age has been observed through the first ten years of life [32]. Growth and atopy are critical health issues for children worldwide; environment, lifestyle, and urbanization changes may profoundly impact both growth and atopy. These trends are an issue because more atopic children are at risk of having improper growth rates [33]. Various factors contribute to growth in children, including underlying medical conditions and psychosocial or environmental problems. However, we did not record each participant’s medication history. Oral medications, especially steroids, may potentially influence children’s growth conditions [34,35]. From an epidemiological perspective, inadequate nutrition is the main cause of poor growth in preschool and school-aged children [36].

Serum IgE is responsible for activating mast cells, which can release histamines, cytokines, growth factors, and chemokines such as IL-4 and IL-13 [37]. Higher interleukin-6, leptin, and TNF-α levels have also been observed in obese people [38]. These chemokines may downregulate T cells. Of the aforementioned studies, some have investigated that a possible connection between atopy and obesity could result in increased systemic inflammation in certain children. While many studies have demonstrated a direct correlation between obesity and atopy [39], some studies have shown the opposite, suggesting that childhood obesity may be associated with a decreased risk of AD and AR in school-aged children [40]. An inverse relationship was noted between central obesity and AR [41]. Increased leptin, a hormone secreted by adipocytes in correlation with total body fat mass and decreased adiponectin levels in obese subjects, play a direct role in regulating inflammation in asthmatics [42,43]. Similarly, leptin augments alveolar macrophage leukotriene synthesis in the rat model [44]. In summary, atopy and growth are likely connected in a multifactorial manner.

In this study, we showed that the positive relationship between BH/BW and serum total IgE levels decreased as children grew older. This finding may be the result of the significant influence of prenatal and early infant diets. Furthermore, our findings regarding the relationship between total IgE and BH/BW/BMI are generally consistent with those in the previous literature. The development of children is multi-dimensional. The internal genetic factor may be the most important in early development; however, while growing up, there are additional developmental influences, such as environmental or social factors [45]. On the other hand, exposure to developmental risks during the earlier years had a higher susceptibility to having adverse effects on growth. One study from China showed that air pollution had more impact on infants than children [46]. In addition, the fact that irritants have a more significant effect on preschool children than their elders may be due to preschoolers’ immature immune systems and narrower airways [47].

Specific IgE testing can improve the diagnostic rate, which may potentially decrease the side effects caused by the skin patch test or by oral food challenges. While clinical history plays a crucial role in establishing an allergy diagnosis, a clinician determines whether to order an IgE test in order to diagnose allergic diseases. In our study, subjects with relevant reported symptoms were selected pursuant to an atopy diagnosis established during outpatient department visits. Selection bias may have been caused by subjects whose BHs were not measured, so the BH data set is lacking some information. Furthermore, we did not record such longitudinal data as serial BW, BH, or mid-parental height. We performed a cross-sectional survey to distinguish and remove the effects of age on the cohorts. The role of age and its influence on BH and BW is also considered in this study. After controlling for age and BW, the IgE levels still showed a significant positive association with BH (*p* < 0.05).

In general, individuals with higher levels of antibodies are more likely to experience allergic symptoms. However, that alone may not be a precise enough indicator for the identification of an atopic condition. The sensitivity and specificity of immunoassays for allergens were 60–90% and 30–95%, respectively [48]. Although the presence of IgE indicates sensitization, its presence does not necessarily indicate allergic disease, and some patients who develop symptoms to an allergen may not necessarily have any allergen-specific IgE during a blood exam.

## 5. Conclusions

High IgE levels were not associated with BH or BW in preschool children.

## Figures and Tables

**Table 1 children-09-00661-t001:** Distribution of age, sex, total serum IgE levels, and 36 allergen-specific IgE.

Characteristic		Case Number (%)
Age (years)	0–6 6–12 12–18	623 (62.7%) 269 (27.1%) 101 (10.2%)
Sex	Male Female	598 (60.2%) 395 (39.8%)
Total IgE levels (kU/L)		443 (median) (range: 2–9756)
Allergen	Latex	14 (1.4%)
	Avocado	14 (1.4%)
	Pork	57 (5.7%)
	Beef	118 (11.9%)
	Milk	11 (1.1%)
	Cheese	24 (2.4%)
	Shrimp	144 (14.5%)
	Crab	120 (12.1%)
	Clam	80 (8.1%)
	Codfish	25 (2.5%)
	Tuna	18 (1.8%)
	Peanut	58 (5.8%)
	Soy	39 (3.9%)
	Wheat	21 (2.1%)
	Brewer’s yeast	14 (1.4%)
	Egg yolk	16 (1.6%)
	Egg white	65 (6.5%)
	Latex	14 (1.4%)
	Chicken feathers	9 (0.9%)
	Bermuda grass	18 (1.8%)
	Black willow	35 (3.5%)
	Eucalyptus	8 (0.8%)
	Japanese cedar	19 (1.9%)
	White mulberry	6 (0.6%)
	Pigweed	49 (4.9%)
	Ragweed mixed	6 (0.6%)
	Timothy grass	9 (0.9%)
	Alternaria	18 (1.8%)
	Aspergillus	9 (0.9%)
	Cladosporium	13 (1.3%)
	Penicillium	9 (0.9%)
	Cat	44 (4.4%)
	Dog	57 (5.7%)
	House dust	273 (27.5%)
	Cockroach mix	212 (21.3%)
Dust mites	Dermatophagoides farina Dermatophagoides pteronyssinus	500 (50.3%) 459 (46.2%)

**Table 2 children-09-00661-t002:** The associations between serum total immunoglobulin (Ig) E levels, body weight, body height, and Body Mass Index (BMI).

IgE Level	Correlation Coefficient Value (*r*)	*p*-Value
All groups (N = 993)		
Body height (N = 581)	0.145	<0.001 *
Body weight (N = 993)	0.086	0.007 *
BMI (N = 581)	0.046	0.267
Preschool group (Age 0–6) (N = 623)		
Body height (N = 348)	0.139	0.009 *
Body weight (N = 623)	0.085	0.034 *
BMI (N = 348)	−0.035	0.515
Elementary school group (Age 6–12) (N = 269)		
Body height (N = 168)	−0.005	0.946
Body weight (N = 269)	−0.008	0.894
BMI (N = 168)	0.002	0.977
High school group (Age 12–18) (N = 623)		
Body height (N = 65)	−0.46	0.715
Body weight (N = 623)	−0.042	0.680
BMI (N = 65)	0.004	0.974
Boys with an age ≥ 11 years old		
Body height (N = 61)	0.003	0.98
Body weight (N = 101)	0.130	0.19
BMI (N = 61)	0.136	0.29
Boys with an age < 11 years old		
Body height (N = 286)	−0.116	0.05
Body weight (N = 497)	−0.060	0.18
BMI (N = 286)	−0.039	0.51
Girls with an age ≥ 10 years old		
Body height (N = 50)	0.154	0.29
Body weight (N = 69)	−0.009	0.94
BMI (N = 50)	−0.054	0.71
Girls with an age < 10 years old		
Body height (N = 184)	0.048	0.52
Body weight (N = 326)	0.018	0.74
BMI (N = 184)	0.085	0.25

N, number. *statistically significant results (*p* < 0.05).

**Table 3 children-09-00661-t003:** The associations between body height and allergen-specific IgE.

Body Height	Correlation Coefficient Value (*r*)	*p*-Value
All groups (N = 581)		
Pork	0.365	<0.001
Egg white	−0.208	<0.001
House dust	0.354	<0.001
Cockroach mix	0.304	<0.001
Dermatophagoides farina	0.368	<0.001
Dermatophagoides pteronyssinus	0.365	<0.001
Preschool group (Age 0–6) (N = 348)		
Pork	0.357	<0.001
House dust	0.314	<0.001
Cockroach mix	0.253	<0.001
Dermatophagoides farina	0.351	<0.001
Dermatophagoides pteronyssinus	0.357	<0.001

**Table 4 children-09-00661-t004:** The associations between body weight and allergen-specific IgE.

Body Weight	Correlation Coefficient Value (*r*)	*p*-Value
All groups (N = 993)		
Pork	0.348	<0.001
House dust	0.307	<0.001
Cockroach mix	0.280	<0.001
Dermatophagoides farina	0.337	<0.001
Dermatophagoides pteronyssinus	0.348	<0.001
Preschool group (Age 0–6) (N = 623)		
House dust	0.274	<0.001
Cockroach mix	0.215	<0.001
Dermatophagoides farina	0.321	<0.001
Dermatophagoides pteronyssinus	0.330	<0.001

## Data Availability

Data are available upon reasonable request.

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
