# Peer review of "Association between Serum Total and Specific Immunoglobulin E Levels and Body Height: A Cross-Sectional Study of Children and Adolescents"

_children, 2022, doi:10.3390/children9050661_

Round 1

Reviewer 1 Report

There needs to be an alignment between the title, hypothesis, results and discussion as per the specific comments. More importantly, the central concept  that forms the basis of the association between allergy and  poor growth) has not been clearly articulated. There is only 1 line hinting to sleep disturbance, without further elaboration- albeit  three references (13-15)  that address this aspect are listed.  

This should form the linking thread across the entire article.

Some grammar and writing style also needs attention. 

Author Response

In response to the Reviewer #1’s specific comments:  

There needs to be an alignment between the title, hypothesis, results and discussion as per the specific comments. More importantly, the central concept that forms the basis of the association between allergy and poor growth has not been clearly articulated. There is only 1 line hinting to sleep disturbance, without further elaboration- albeit three references (13-15) that address this aspect are listed. 

Response

  • Thank you for the great comments.
  • According to the main text, we aimed to investigate the association between total, specific IgE and body height/weight/ body mass index (BMI); but not "growth". The latter would require longitudinal data from a cohort study but this is a cross-sectional study. We corrected the usage of growth to more precise body height/weight/body mass index.
  • We put more emphasis on the cross-sectional study in the topic and manuscript.
  • We added the reference to place much more emphasis on the connection between atopy and growth in a multifactorial manner. Allergic diseases and growth, which interrelate reciprocally, are a great concern.

Sung WH, Yeh KW, Huang JL, Su KW, Chen KF, Wu CC, et al. Longitudinal changes in body mass index Z-scores during infancy and risk of childhood allergies. Journal of microbiology, immunology, and infection = Wei mian yu gan ran za zhi. 2021.

Revised manuscript

  • Association between serum total and specific immunoglobulin E levels and body height: a cross-sectional study of children and adolescents. (Topic)
  • The principal research question in this cross-sectional investigation concerned the association between serum total, specific immunoglobulin E (IgE) levels, and body height (BH)/weight (BW)/body mass index (BMI). (Abstract)
  • The results of this study indicated that the level of total IgE was not associated with BH in the preschool age group. (Abstract)
  • In this cross-sectional study, we aimed to investigate the association between total, specific IgE and BH/body weight (BW)/body mass index (BMI). We compared total and specific IgE levels among allergic children with different BH/BW/BMI and hypothesized that lower BH was not associated with higher IgE levels. The next research question focused on the effect of age of allergic children on the association. (Introduction)
  • Participants and study design
    • The participants enrolled in our cross-sectional study were children with a diagnosis of current atopic status including AD, AR, and/or asthma by a doctor in the outpatient pediatrics department at Kaohsiung Chang Gung Memorial Hospital in Taiwan.
  • The results of this study were adjusted by average pubertal age and showed no significant effect of pubertal status on total IgE and BH/BW/BMI. (Results)
  • The impact of BH was similar to the BH survey in the young adults with another common IgE mediated food allergy, cow milk. (Discussion)
  • It may be because the diet control before school is more strict, which affects nutrient intake. (Discussion)
  • Furthermore, our findings regarding the relationship of total IgE with BH/BW/BMI are generally consistent with those of the previous literature. (Discussion)
  • High IgE levels were not associated with BH or BW in preschool children. (Conclusion)
  • Furthermore, we did not record such longitudinal data as serial BW, BH, or mid-parental height. (Discussion)

Reviewer 2 Report

Thank you for giving me the opportunity to read this manuscript of results from a very interesting study. I have some commments.

  1. Title:  Following the STROBE guidelines for reporting results from observational studies, the title should include the information of the study design, e.g. cross-sectional study (or cross-sectional analysis).
  2. Research question(s):  According to the abstract, the authors aimed to examine the correlation between „these factors“ i.e., growth and allergies and serum total and specific immunoglobulin E (IgE) levels. However, according to the main text, they aimed to investigate the association between total, specific IgE and body height/weight/ body mass index (BMI); but not "growth". The latter would require longitudinal data but this is a cross-sectional study I assume. In their main text, the authors further specify in their objective that total and specific IgE levels are only examined in allergic children. This (purely allergic population) should also be made clearer throughout the manuscript, since some non-allergic children also habe elevated IgE concentrations.   I think the authors should be more precise with their primary and possibly secondary research questions in the main text; and make sure that the abstract inlcudes a precisely worded main research question, in accordance to the objectives in the main text.
  3. Abstract, Results: The authors should add information on the effect size of the association between total IgE levels with body height and body weight in the preschool age group and report not only the p-values.
  4. Abstract, results: This should be revised according to my suggestions number 11-13 (see below).
  5. Abstract, Conclusions: I don’t understand the expression „growth of body height“? Especially in a cross-sectional study like this one, it is not possible to examine ‚growth‘.
  6. Introduction, first sentence: The statement of the first sentence is not true and selectively refers to some almost 30 year old references. Occurrence of allergies have increased but not all over the world. In some areas this was true, in others allergies have reached a plateau, and in some areas their occurrence decreased, especially where it was very high at first (Great Britain etc). The best and most representative data on these divergent time trends comes from the ISAAC project (by a team from Auckland, New Zealand), the largest study with almost 500,000 children worldwide (see Asher MI et al, The Lancet 2006, Worldwide time trends in the prevalence of symptoms of asthma, allergic rhinoconjunctivitis, and eczema in childhood …)
  7. Introduction: for prevalence or incidence, a maximum of one decimal is sufficient and makes it easier for the readers to understand the text. Throughout the manuscript, I would avoid decimals in reporting such high percentages for the occurence of allergies.
  8. Materials and Methods: I suggest to structure this section according to the STROBE guidelines, and include the precise information under the different subheadings.
  9. Material and Methods, Statistical methods: Was there a sample size calculation, to assess if the sample was sufficinet for detecting clinically relevant differences? Please report how missing data was handled. Please report what type of multiple regression analysis was conducted; stepwise regression, with e.g., backward or forward selection of variables? Please report how potential confounders were selected and how (as continuous or categorical variables) included in the model.  Please also report how the problem of ‚multiple testing‘ was handled in this analysis, and if not, justify why there was no control for ‚multiple testing‘ (please consult a statistician or experienced epidemiologist).
  10. Results: Follow STROBE here as well please.
  11. Results: the authors need to consult a statistician to interpret and to report the results correctly. For instance, „Pearson’s correlation= 0.139,…“ is not correct, the value indicates a coefficient, reported as r. It should be: coefficient or just r = 0.139 etc. This also needs to be corrected throughout the manuscript including the tables.
  12. Results: I think that the follwoing sentence is not helpful for the reader: „There were significant positive association between total IgE levels with body height (N=581, p<0.001) and body weight (N=993, p=0.007) across all age groups.“ This statement by the authors is only based on the p-values and should be corrected to „There were statisctically significant associations..“. However, looking at the coefficients of r=0.145 and  r= 0.086 I don’t see a (clinically) relevant correlation (association).   Values of r between 0 and 0.2 should be interpreted as „no correlation“ (no association), regardless of the corresponding p-value, which is just a measure of how certain we are about the r coefficient! Further, values of >0.2 to 0.4 may be interpreted as weak correlations (associations), r >0.4 – 0.6 as moderate correlation, and r >0.6 to 1 as good or very good/excellence (the latter if >0.8). Please consult a statistican for help with this.  Looking at tables 2-4, I see no moderate or good correlation (association), only very few weak ones with coefficients of around 0.3. This should be emphasized in teh description and interpretation of the results, including the conclusions of the main text and abstract. The current conlucions of the main text are not fully supported by own data.
  13. I strongly recommend that the text of the Results and Discussion section should be rewritten considering in a fisrt step the correlation coeffecients; and only in a second step look at the p-values of the coefficients, if they are higher than 0.2. Those below 0.2 are not (clinially) relevant for the research questions that the authors aimed to examine, and don’t need to be stretched. Overall, the findings from this study do not add evidence that height or weight are related to total or specific IgE levels at any age in childhood. I think this is a very important finding considering our knowledge gaps. The study is very relevant in that sense. It will add to our scientific knowledge, if the results are interpreted more critical. This should be reflected in the conslusions.

I thank you again for the opportunity to review this manuscript and hope that my comments are helpful.

Author Response

In response to the Reviewer #2’s specific comments:

Title: Following the STROBE guidelines for reporting results from observational studies, the title should include the information of the study design, e.g. cross-sectional study (or cross-sectional analysis).

Response

  • According to the reviewer’s suggestions, we have added the study design for the title.

Revised manuscript

  • Association between serum total and specific immunoglobulin E levels and body height: a cross-sectional study of children and adolescents. (Topic)

Research question(s): According to the abstract, the authors aimed to examine the correlation between, these factors“ i.e., growth and allergies and serum total and specific immunoglobulin E (IgE) levels. However, according to the main text, they aimed to investigate the association between total, specific IgE and body height/weight/ body mass index (BMI); but not "growth". The latter would require longitudinal data but this is a cross-sectional study I assume.

Response

  • We agree with the reviewer that the use of the specific term ”growth” should be more precise.

Revised manuscript

  • The results of this study indicated that the level of total IgE was not associated with BH in the preschool age group. (Abstract)
  • The results of this study were adjusted by average pubertal age and showed no significant effect of pubertal status on total IgE and BH/BW/BMI. (Results)
  • The impact of BH was similar to the BH survey in the young adults with another common IgE mediated food allergy, cow milk. (Discussion)
  • It may be because the diet control before school is more strict, which affects nutrient intake. (Discussion)
  • Furthermore, our findings regarding the relationship of total IgE with BH/BW/BMI are generally consistent with those of the previous literature. (Discussion)
  • High IgE levels were not associated with BH or BW of preschool children. (Conclusion)

In their main text, the authors further specify in their objective that total and specific IgE levels are only examined in allergic children. This (purely allergic population) should also be made clearer throughout the manuscript since some non-allergic children also have elevated IgE concentrations.

Response

  • The definition of the population includes purely allergic population made clearer throughout the manuscript as suggested.

Revised manuscript

  • We compared total and specific IgE levels among allergic children with different BH/BW/BMI and hypothesized that lower BH was not associated with higher IgE levels. (Introduction)
  • The participants enrolled in our cross-sectional study were children with a diagnosis of current atopic status including AD, AR, and/or asthma by a doctor in the outpatient pediatrics department at Kaohsiung Chang Gung Memorial Hospital in Taiwan. (Methods)
  • We identified BH had a positive association with Dermatophagoides farina sensitization in the pre-school subgroup of allergic children. (Discussion)

I think the authors should be more precise with their primary and possibly secondary research questions in the main text; and make sure that the abstract includes a precisely worded main research question, in accordance with the objectives in the main text.

Response

  • Thanks to the reviewer for the suggestion. We pointed out the research questions in the abstract and main text.

Revised manuscript

  • The principal research question in this cross-sectional investigation concerned the association between serum total, specific immunoglobulin E (IgE) levels, and body height (BH)/weight (BW)/body mass index (BMI). (Abstract)
  • In this cross-sectional study, we aimed to investigate the association between total, specific IgE and BH/body weight (BW)/body mass index (BMI). We compared total and specific IgE levels among allergic children with different BH/BW/BMI and hypothesized that lower BH was not associated with higher IgE levels. The next research question focused on the effect of age of allergic children on the association. (Introduction)

Abstract, Results: The authors should add information on the effect size of the association between total IgE levels with body height and body weight in the preschool age group and report not only the p-values.

Revised manuscript

  • There was a statistically significant positive association between total IgE levels with BH (N=348) and BW (N=623) in the preschool age group (<6 years old, p-values of 0.009, 0.034, respectively). (Abstract)

Introduction, first sentence: The statement of the first sentence is not true and selectively refers to some almost 30 year old references. Occurrence of allergies have increased but not all over the world. In some areas this was true, in others allergies have reached a plateau, and in some areas their occurrence decreased, especially where it was very high at first (Great Britain etc). The best and most representative data on these divergent time trends comes from the ISAAC project (by a team from Auckland, New Zealand), the largest study with almost 500,000 children worldwide (see Asher MI et al, The Lancet 2006, Worldwide time trends in the prevalence of symptoms of asthma, allergic rhinoconjunctivitis, and eczema in childhood …)

Response

  • We added references as suggested.

Asher MI, Montefort S, Björkstén B, Lai CK, Strachan DP, Weiland SK, et al. Worldwide time trends in the prevalence of symptoms of asthma, allergic rhinoconjunctivitis, and eczema in childhood: ISAAC Phases One and Three repeat multicountry cross-sectional surveys. Lancet (London, England). 2006;368(9537):733-43.

Revised manuscript

  • The atopy rate in children has increased significantly in some areas, particularly in industrialized countries. The most representative data on these divergent time trends comes from the ISAAC project showing increased prevalence of allergic rhinoconjunctivitis and eczema symptoms for the 6–7 and 13–14 year age group in Taiwan after seven years of the first survey between 2002 and 2003. (Introduction)

Introduction: for prevalence or incidence, a maximum of one decimal is sufficient and makes it easier for the readers to understand the text. Throughout the manuscript, I would avoid decimals in reporting such high percentages for the occurrence of allergies.

Response

  • Following the suggestion, we use one decimal number for prevalence or incidence.

Revised manuscript

  • According to data from studies done in Taiwan, the prevalence of childhood asthma increased from 1.3% in 1974 to 5.1% in just ten years. A similar study in Taiwan demonstrated that atopy has continued to increase in recent decades. For example, the prevalence of asthma more than tripled from 1987 to 2002, jumping from 2.2% to 7.0%. Such increases were also noted in allergic rhinitis (AR, from 5.1% in 1987 to 27.6% in 2002) and atopic dermatitis (AD, from 1.1% in 1987 to 3.4% in 2002). (Introduction)

Materials and Methods: I suggest structuring this section according to the STROBE guidelines, and including the precise information under the different subheadings.

Response

  • Thank you for the valuable recommendation. We followed the STROBE checklist showing participants, study design, setting, variables, data sources/ measurement, bias, study size, quantitative variables, and statistical methods and reorganized it with appropriate subheadings in Methods.

Revised manuscript

  • Participants and study design
    • The participants enrolled in our cross-sectional study were children with a diagnosis of current atopic status including AD, AR, and/or asthma by a doctor in the outpatient pediatrics department at Kaohsiung Chang Gung Memorial Hospital in Taiwan. We enrolled a total of 993 subjects aged from 6 months to 18 years during December 2015 - August 2016 to determine the relationship between IgE levels and body size including BH, BW, and BMI. We divided data into groups according to the definition of precocious puberty age in Taiwan of boys (11years old) and girls (10 years old) for consideration of the pubertal effect. Children with an underlying medical condition or congenital abnormality were excluded, as were subjects with heart, thyroid, chronic lung, renal, or liver disease. Informed consent was waived and all methods carried out in accordance with relevant guidelines were approved by the Institutional Review Board of Chang Gung Medical Foundation (IRB No: 106-0014D).
  • Assessments
    • A complete panel of 36 allergen-specific IgE was taken from each participant's peripheral blood by using the MAST allergen test (the multiple allergosorbent test system, chemiluminescent immunoassay sandwich method), BW (kilogram), BH (centimeter) and total IgE data. Of the 993 participants, 581 (58.5%) had a BH measured using appropriate measuring devices. The missing data for BH was not included in the analysis related to BH or BMI. The study size provided the power greater than 90%. Standing BH was measured without shoes. The serum total IgE levels and 36 allergen-specific IgE antibodies against latex, chicken feathers, Bermuda grass, black willow, eucalyptus, Japanese cedar, white mulberry, pigweed, ragweed mix, timothy grass, alternaria, aspergillus, cladosporium, penicillium, cat, dog, house dust, cockroach mix, Dermatophagoides farina, Dermatophagoides pteronyssinus, avocado, pork, beef, milk, cheese, shrimp, crab, clam, codfish, tuna, peanut, soy, wheat, brewer’s yeast, egg yolk, and egg white were analyzed with BH and BW. The growth status was evaluated according to "Child growth standards" of the World Health Organization (WHO) (https://www.who.int/toolkits/ child-growth-standards) and (https://growth.healthinfo.tw/).
  • Statistical methods
    • We used the statistical software SPSS (Version 13.0) to perform all analyses. Participants were categorized into three different age groups, including preschool (<6 years old), elementary school (6-12 years old), and high school (12-18 years old). We evaluated the statistical significance between body size and total IgE levels using Pearson’s correlation; body size and allergen-specific IgE (grade zero to four) using Spearman’s correlation. A p-value less than 0.05 was considered statistically significant for the main effects. We determined the relationships between specific IgE and body size by applying a general linear model followed by a backward selection of variables with age, sex, specific, IgE, and total IgE because BH increased with age and age, total IgE, and sex had significant associations with allergen sensitization. Age and total IgE were considered continuous variables. Sex and specific IgE were considered categorical variables. Grade zero was called negative sensitization. Grade one to four referred to positive sensitization. Then, false discovery rate correction for multiple comparisons was conducted. The statistical power was calculated in the general linear model.
  • Patient and public involvement
    • Patients and/or the public were not involved in the design, conduct, or dissemination plans of this research.

Material and Methods, Statistical methods: Was there a sample size calculation, to assess if the sample was sufficient for detecting clinically relevant differences?

Response

  • The power of the model showed the sample was sufficient for detecting clinically relevant differences.

Revised manuscript

  • The study size provided the power greater than 90%. (Methods)
  • The statistical power was calculated in the general linear model. (Methods)
  • Of the 36 kinds of specific IgE, egg whites demonstrated a significant negative association with BH in subjects under 6 years old after performing a general linear model (p=0.009) with power greater than 90%. (Results)

Please report how missing data were handled.

Revised manuscript

  • The missing data for BH was not included in the analysis related to BH or BMI. (Methods)

Please report what type of multiple regression analysis was conducted; stepwise regression, with e.g., backward or forward selection of variables? Please report how potential confounders were selected and how (as continuous or categorical variables) were included in the model. Please also report how the problem of‚ multiple testing‘ was handled in this analysis, and if not, justify why there was no control for ‚multiple testing‘ (please consult a statistician or experienced epidemiologist).

Response

  • After consulting our statistician team, we determined the relationships between specific IgE and body size by applying a general linear model followed by a backward selection. Then, false discovery rate correction for multiple comparisons was conducted.

Revised manuscript

  • We determined the relationships between specific IgE and body size by applying a general linear model followed by a backward selection of variables with age, sex, specific, IgE, and total IgE because BH increased with age and age, total IgE, and sex had significant associations with allergen sensitization. Age and total IgE were considered continuous variables. Sex and specific IgE were considered categorical variables. Grade zero was called negative sensitization. Grade one to four referred to positive sensitization. Then, false discovery rate correction for multiple comparisons was conducted. The statistical power was calculated in the general linear model. (Methods)
  • Acknowledgements
    • We would like to thank the Biostatistics Center, Kaohsiung Chang Gung Memorial Hospital, for statistics work.

Results: the authors need to consult a statistician to interpret and to report the results correctly. For instance, „Pearson’s correlation= 0.139,…“ is not correct, the value indicates a coefficient, reported as r. It should be: coefficient or just r = 0.139 etc. This also needs to be corrected throughout the manuscript including the tables.

Response

  • Thank you for the correction. We have revised results and tables 2 and 3 accordingly.

Revised manuscript

  • When divided into the different age groups, statistically significant positive associations between BH (Pearson’s correlation coefficient r=0.139, N=348, p= 0.009) and BW (Pearson’s correlation coefficient r=0.085, N=623, p=0.034) with total IgE were found only in the preschool age group (Table 2). However, looking at the coefficients of r < 0.2, we did not find a clinically relevant association in Table 2. (Results)

Results: I think that the following sentence is not helpful for the reader: There were significant positive associations between total IgE levels with body height (N=581, p<0.001) and body weight (N=993, p=0.007) across all age groups.“ This statement by the authors is only based on the p-values and should be corrected to “There were statistically significant associations..“. However, looking at the coefficients of r=0.145 and r= 0.086 I don’t see a (clinically) relevant correlation (association). Values of r between 0 and 0.2 should be interpreted as „no correlation“ (no association), regardless of the corresponding p-value, which is just a measure of how certain we are about the r coefficient! Further, values of >0.2 to 0.4 may be interpreted as weak correlations (associations), r >0.4 – 0.6 as moderate correlation, and r >0.6 to 1 as good or very good/excellence (the latter if >0.8). Please consult a statistician for help with this.  Looking at tables 2-4, I see no moderate or good correlation (association), only very few weak ones with coefficients of around 0.3. This should be emphasized in the description and interpretation of the results, including the conclusions of the main text and abstract. The current conclusions of the main text are not fully supported by own data.

Response

  • Thank you for the great recommendation. We emphasized coefficients r > 0.2 in Tables 3 and 4. We revised the manuscript accordingly after consulting our statistician team.

Revised manuscript

  • There was a statistically significant positive association between total IgE levels with BH (N=348) and BW (N=623) in the preschool age group (<6 years old, p-values of 0.009, 0.034, respectively). (Abstract)
  • When divided into the different age groups, statistically significant positive associations between BH (Pearson’s correlation coefficient r=0.139, N=348, p= 0.009) and BW (Pearson’s correlation coefficient r=0.085, N=623, p=0.034) with total IgE were found only in the preschool age group (Table 2). However, looking at the coefficients of r < 0.2, we did not find a clinically relevant association in Table 2. (Results)
  • Table 3/4 showed significant associations with p<0.05 and coefficients r >0.2 or <-0.2 between specific IgE and BH/BW. (Results)
  • We identified BH had a positive association with Dermatophagoides farina sensitization in the pre-school subgroup of allergic children. The positive association between Dermatophagoides farina sensitization and BH was possibly due to increased surface area in contact with allergens. (Discussion)
  • Conclusion
    • High IgE levels were not associated with BH or BW in preschool children.
  • Acknowledgements
    • We would like to thank the Biostatistics Center, Kaohsiung Chang Gung Memorial Hospital, for statistics work.

This manuscript is a resubmission of an earlier submission. The following is a list of the peer review reports and author responses from that submission.

Round 1

Reviewer 1 Report

Although the concept of correlating IgE levels with growth in a sizable cohort of children and adolescents is interesting, there are significant flaws in the analysis and presentation of the results. It is not clear to the reader what was the outcome of the study: Specific IgEs for specific allergens? Total IgE levels? and compared to what? Height, weight, BMI? in addition BMI are expressed either in 95th %tiles or z-scores. Was multiple regression analysis conducted for all variables examined? Finally, although a lack of complete data is mentioned in the discussion, there is no further information provided.  

Reviewer 2 Report

The measurement of total IgE is not enough important for diagnosis of an allergy disease.  More relevant and key for diagnosi is the measurement of specific IgE. Height and weight are not important factor for the development and severity of an allergy disease.

In your title, you do not mention the measurement of specific IgE.

You need to add references (as I ask in the file)

I do not understand the relevance of this study.
